**Data Availability Statement:** All relevant data are within the manuscript and its Supporting Information files.

**Funding:** JMH: Eastern Norway Regional Health Authority, 2303 Hamar. The funders had no role in

# Left ventricular dysfunction in COPD without pulmonary hypertension

Janne M. Hilde[1,2], Jonny Hisdal[1,3], Ingunn Skjørten[2,4], Viggo Hansteen[1], Morten N. Melsom[4], Ole J. Grøtta[5], Milada C. Småstuen[6], Ingebjørg Seljeflot[2,7], Harald Arnesen[2,7], Sjur Humerfelt[4], Kjetil Steine [2,8]*

**1** Department of Cardiology, Oslo University Hospital-Aker, Oslo, Norway, **2** Faculty of Medicine, University of Oslo, Oslo, Norway, **3** Section of Vascular Investigations, Oslo University Hospital-Aker, Oslo, Norway, **4** Department of Pulmonary Medicine, Oslo University Hospital-Aker, Oslo, Norway, **5** Department of Radiology, Oslo University Hospital-Aker, Oslo, Norway, **6** College of Applied Sciences, Fac. of Health Sciences, Oslo, Norway, **7** Center for Clinical Heart Research, Department of Cardiology, Oslo University Hospital- Ullevål, Oslo, Norway, **8** Department of Cardiology, Akershus University Hospital, Oslo, Norway

* kjetil.steine@medisin.uio.no

## Abstract

### Objectives

We aimed to assess prevalence of left ventricular (LV) systolic and diastolic function in stable cohort of COPD patients, where LV disease had been thoroughly excluded in advance.

### Methods

100 COPD outpatients in GOLD II-IV and 34 controls were included. Patients were divided by invasive mean pulmonary artery pressure (mPAP) in COPD-PH ($\geq$25 mmHg) and COPD-non-PH (<25 mmHg), which was subdivided in mPAP $\leq$20 mmHg and 21–24 mmHg. LV myocardial performance index (LV MPI) and strain by tissue Doppler imaging (TDI) were used for evaluation of LV global and systolic function, respectively. LV MPI $\geq$0.51 and strain $\leq$-15.8% were considered abnormal. LV diastolic function was assessed by the ratio between peak early (E) and late (A) velocity, early TDI E´, E/E´, isovolumic relaxation time, and left atrium volume.

### Results

LV MPI $\geq$0.51 was found in 64.9% and 88.5% and LV strain $\leq$-15.8% in 62.2.% and 76.9% in the COPD-non-PH and COPD-PH patients, respectively. Similarly, LV MPI and LV strain were impaired even in patients with mPAP <20 mmHg. In multiple regression analyses, residual volume and stroke volume were best associated to LV MPI and LV strain, respectively. Except for isovolumic relaxation time, standard diastolic echo indices as E/A, E´, E/E´ and left atrium volume did not change from normal individuals to COPD-non-PH.

study design, data collection and analysis, decision to publish, or preparation of the manuscript.

**Competing interests:** The authors have declared that no competing interests exist.

## Conclusions

Subclinical LV systolic dysfunction was a frequent finding in this cohort of COPD patients, even in those with normal pulmonary artery pressure. Evidence of LV diastolic dysfunction was hardly present as measured by conventional echo indices.

## Introduction

In patients with chronic obstructive pulmonary disease (COPD), and in particular in those with severe emphysema, pulmonary hypertension and right ventricular (RV) enlargement, the left ventricle is compressed. However, for the majority of these patients, the left ventricular (LV) ejection fraction is within normal limits [1]. The majority of COPD patients with such findings shows a leftward ventricular septum deviation, most marked at end systole and early diastole, associated with a distortion of LV geometry and reduction of early diastolic filling [1, 2]. This mechanism is thought to be the most likely cause for the decreased LV size, stroke volume and thus under filling of the left ventricle [1–3]. Moreover, Barr RG et al. demonstrated in a population based study of 2816 participants that increasing extent of emphysema on CT scanning and more severe airflow obstruction were linearly related to impaired left ventricular filling and reduced stroke volume without changes in LV ejection fraction [4]. However, LV ejection fraction by two-dimensional echocardiography can be normal despite LV dysfunction [5]. Because of this and the symptom similarity between heart failure and COPD, there is a need for more sensitive diagnostic tools to unmask if LV systolic subclinical impairment is present in COPD patients. Moreover, there is only a few existing studies on COPD and LV function recruiting mostly severe COPD, having small number of participants, and they are also deficient with respect to invasive pressure data [6–8].

Tissue Doppler imaging (TDI) is a more sensitive tool to detect subclinical systolic LV dysfunction. The present group has previously demonstrated reduced RV function by TDI, even in COPD patients without pulmonary hypertension [9]. To the best of our knowledge, there are only two small studies on COPD and LV function using TDI [10, 11]. However, none of these two studies had measured pulmonary pressure by right heart catheterization.

The aim of the present study was therefore to characterize LV function and investigate the prevalence of subclinical LV dysfunction by these novel imaging techniques in a patient population of stable COPD patients with and without pulmonary hypertension and without known LV disease. We hypothesized that even in such a cohort, geometric alterations in structure induces LV systolic and diastolic dysfunction in the majority of the patients.

## Methods

### Study population

Hundred outpatients with stable COPD of different severity and free of clinical cardiovascular disease, were consecutively included from 2006 to 2010 (Table 1). Baseline characteristics of the study population have previously been described in details [9]. The diagnosis of COPD was based on a history of cigarette smoking with at least 10 pack-years and spirometry irreversible airway obstruction according to current guidelines [12]. Spirometry, body plethysmography, diffusion capacity for carbon monoxide, and arterial blood gases were performed according to guidelines. Prediction equations for forced expiratory volume in the first second, forced expiratory capacity and static lung volumes were used [13, 14]. Patients were classified

**Table 1. Clinical characteristic of controls and patients with chronic obstructive pulmonary disease without (COPD-non-PH) and with pulmonary hypertension (COPD-PH).**

| Variables (unit) | Controls (n = 34) | COPD-non-PH (n = 74) | COPD-PH (n = 26) |
|---|---|---|---|
| **Demographics** | | | |
| Age (years) | 63±7 | 64±6 | 62±8 |
| Body mass index (kg/m$^2$) | 25±3 | 24±4 | 25±7 |
| Body surface area (m$^2$) | 1.9 ±0.2 | 1.8±0.2 | 1.8±0.3 |
| Systolic blood pressure (mmHg) | 120±17 | 139±22* | 140±18* |
| Diastolic blood pressure (mmHg) | 76±12 | 69±11* | 68±13* |
| Heart rate (bpm) | 68±10 | 69±13 | 79±15*[†] |
| Pack-years of smoking | 8±9 | 40±19* | 39±20* |
| **Categorical variables** | n or n/% | n or n/% | n or n/% |
| Sex Female/Male (n) | 19/15 | 36/38 | 15/11 |
| Hypertension (n/%) | 3/9 | 22/30 | 11/42*[†] |
| Diabetes (n/%) | 0/0 | 5/7 | 4/15 |
| GOLD stages I/II/III/IV(n) | 0 | 2/35/23/14 | 0/2/8/16 |
| Smoking habits[‡] (n) | 2/13/19 | 51/23/0* | 18/8/0* |
| **Pulmonary function and art. blood gases** | | | |
| FEV$_1$% predicted | 98±10 | 47±16* | 32±12*[†] |
| FVC % predicted | 105±11 | 77±20* | 61±16*[†] |
| FEV$_1$/FVC (%) | 76±4 | 50±11* | 43±13*[†] |
| Total lung capacity % predicted | 100±15 | 122±20* | 133±25* |
| Residual volume % predicted | 119±16 | 191±59* | 241±60*[†] |
| DLCO % predicted | 100±15 | 58±20* | 36±20*[†] |
| PaO$_2$ (kPa) | - | 9.9±1.2 | 8.2±1.6[†] |
| PaCO$_2$ (kPa) | - | 5.3±0.6 | 5.7±0.8 |
| Biomarkers | Geometric mean | Geometric mean | Geometric mean |
| NT pro-BNP (pmol/l) | | 9.8 (9.1,10.5) | 10.0 (9.1,10.8) |
| CRP (mg/L) | 0.78(0.40, 2.04) | 3.97(2.11,7.64) * | 4.10(1.29, 5.07) * |
| White blood cell (x10$^9$/L) | 5.60 (4.58, 6.28) | 7.30 (6.0, 8.85)* | 7.90 (6.30, 9.60)* |
| IL-6 (pg/mL) | 1.81 (1.16, 2.75) | 3.50 (2.17, 5.74)* | 3.71(2.33, 6.53)* |
| TNF α (pg/mL) | 1.30 (1.05, 1.47) | 1.33 (1.10, 1.57) | 1.32 (1.11, 1.60) |

Values are mean±SD. CRP: C-reactive protein, DLCO: Diffusion capacity for carbon monoxide of the lungs, FEV$_1$: Forced expiratory volume in the first second, FVC: Forced vital capacity, GOLD: Global Initiative for Chronic Obstructive Lung Disease, IL: Interleukin, PaO$_2$: Arterial oxygen tension; PaCO$_2$: Arterial carbon dioxide tension, TNF: Tumor necrosis factor.

* Significantly different from controls ($p<0.01$),

[†] significantly different from COPD-non-PH ($p<0.01$).

[‡] Former, current and never smoker, respectively.

according to the criteria of the Global Initiative for Chronic Obstructive Lung Disease [12]. All patients had optimal bronchodilator therapy.

Caucasians, 40–75 years, with spirometry confirmed COPD in Global Initiative for Chronic Obstructive Lung Disease stages I-IV, all current or former smokers were included. They had to be free of COPD exacerbations the last two months prior to inclusion. All participants underwent pre-inclusion screening, including resting ECG and a dynamic exercise test on bicycle ergometer. Patients with history of congenital, rheumatic, valvular and ischemic heart disease, treated arterial hypertension with blood pressure >160/90 mmHg, arrhythmias (including atrial fibrillation), other acute or chronic pulmonary disease, malignancy, hyper- and hypothyroidism, systemic inflammatory diseases and renal failure, were excluded. Patients

using beta-blockers, Warfarin or Clopidogrel were also excluded. Subjects were classified as having diabetes when being under treatment for insulin-dependent or non-insulin dependent diabetes or having fasting blood glucose >7 mmol/L. Use of lipid lowering medication was present in 9.8% of COPD patients.

The COPD patients were compared to an age and gender matched control group (n = 34) and evaluated healthy by clinical, biochemical and imaging investigations. The study complies with the Declaration of Helsinki, and was approved by Regional committee for medical and health research ethics south-east and performed at Oslo university hospital-Aker. Written informed consent was obtained from all the subjects.

## Blood collections

Venous blood samples were collected in fasting condition between 08.00 and 10.00 a.m. and centrifuged at 2500 x g for 15 min. Serum and EDTA plasma were stored at -80 ˚C for subsequent analyses in batch. Routine analyses were performed by conventional methods. Inflammatory markers were determined by ELISA from R&D Systems Europe, (Abingdon, Oxon, UK), except for CRP which was analyzed by ELISA from DRG Instruments (GmbH, Germany). Serum was used for all. Arterial blood gases from radial artery in resting condition was collected and analyzed immediately.

## Echocardiography

All study patients underwent a comprehensive Doppler echocardiographic examination prior to and within 120 minutes of right-heart catheterization [9].

LV internal dimension, septal and posterior wall thickness, and LV mass were measured in end-diastole using the parasternal long-axis view [15, 16]. LV MPI and isovolumic relaxation time were measured by pulsed wave TDI and four-chamber view at the basis of the septal and lateral mitral leaflet and averaged. LV myocardial strain was measured by post-processing at the basal third of the septal and lateral LV walls of the apical four-chamber TDI loop and averaged. An LV MPI ≥0.51, LV Strain ≤15.8%, isovolumic relaxation time ≥87 ms and isovolumic relaxation time adjusted for heart rate ≥99 ms (mean+three standard deviations of the controls for all four) were considered abnormal. Isovolumic relaxation time was adjusted for heart rate by dividing the observed Isovolumic relaxation time with the RR interval in seconds.

The mitral inflow measurements included peak early filling (E) and late diastolic filling (A) velocity and the E/A ratio by pulsed Doppler [17]. Early diastolic pulsed TDI (E′) peak velocity was measured at similar locations as LV MPI and averaged, and E/E′ was calculated as surrogate for LV filling pressure [17]. LV ejection fraction and volumes were calculated by the biplane method (modified Simpson's rule) using apical four and two chamber views [15]. Indexed left atrial volume was calculated using the four and two chamber views at end-systole [15]. Right atrium volume was measured by single-plane area-length algorithm from four-chamber view [18]. RV parameters were obtained as previously reported [9].

Real-time three dimensional echo was used to acquire full-volumetric data sets of LV and RV from four ECG-triggered sub-volumes. Post-processing analysis (RV-Function, TomTec Imaging system GmbH, Unterschleissheim, Germany) was performed with semi-automatic software with predefined LV and RV views for endocardial contours delineation.

## Cardiac magnetic resonance imaging

COPD patients (n = 100) were scanned using a 1.5-T Siemens Advanto (Siemens Medical Systems, Erlangen, Germany) with a phased-array body-coil, and acquisition of two- and four-

chamber localizers was performed as previously described [9]. All CMR data were processed by a blinded observer (OJG).

## Hemodynamic measurements

Right heart catheterization was performed at rest with the patient supine as previously described [9]. Electrocardiogram and heart rate were recorded continuously, and cardiac output was determined by thermodilution technique. Pulmonary vascular resistance and LV filling resistance were calculated [9]. Results were stratified according to presence or absence of PH; COPD-PH if mPAP $\geq$25 (n = 26) and COPD-non-PH if mPAP<25 mmHg (n = 74) by right heart catheterization. Furthermore, patients with mPAP <25mmHg were subdivided into $\leq$20mmHg (n = 53) and 21-24mmHg (n = 21).

## Statistical analyses

Continuous variables were described as mean and standard deviation and categorical variables as counts and (%). Crude differences between controls and COPD subgroups and between two COPD subgroups were assessed using analysis of variance with Bonferroni correction and two-independent samples t-test. Pairs of categorical variables were compared using Chi-squared test. Possible associations between LV global function by LV MPI and LV systolic function by strain versus functional echo and invasive indices for RV function and volumes, lung function and hyperinflation indices and standard risk factor for LV disease were evaluated using linear regression models. Variables that reached p<0.1 in univariate analyses were entered into multiple regression models. P-values <0.05 were considered statistically significant. All analyses were performed using SPSS 21 and SigmaPlot v.12.

# Results

## Demographic and clinical data

Clinical data including biomarkers and hemodynamics in controls and patients are summarized in Tables 1 and 2, respectively. Heart rate was higher in the COPD-PH patients vs. controls and vs. COPD-non-PH. All results, also including LV global, systolic and diastolic function, LV size and mass and left atrial size, are summarized in Table 3.

**Table 2. Right heart catheterization values in patients with chronic obstructive pulmonary disease without (COPD-non-PH) and with pulmonary hypertension (COPD-PH).**

| Variables (unit) | COPD-non-PH (n = 74) | COPD-PH (n = 26) |
| --- | --- | --- |
| Mean pulmonary artery pressure (mmHg) | 18±3 | 29±4* |
| Mean pulmonary wedge pressure (mmHg) | 8±4 | 11±3* |
| Mean right atrial pressure (mmHg) | 5±3 | 7±3* |
| Pulmonary vascular resistance (WU) | 2.0±0.9 | 3.4±1.5* |
| Systemic vascular resistance (WU) | 16±4 | 18±4 |
| Systemic arterial compliance (ml/mmHg) | 1.1±0.4 | 1.0±0.4 |
| Cardiac index (l/min/m$^2$) | 2.9±0.4 | 3.1±0.6* |
| Stroke volume indexed (ml/m$^2$/beat) | 39±7 | 37±6 |
| Heart rate (beats/min) | 73±12 | 83±14* |
| Left ventricular filling resistance (mmHg/L/min) | 1.6±0.7 | 2.0±0.6* |

Values are mean±SD (standard deviation).

* Significantly different from COPD-non-PH.

**Table 3. Echocardiographic measurements of LV global, systolic and diastolic function, and dimensions of the LV, left atrium and right atrium in healthy controls, patients with chronic obstructive pulmonary disease without (COPD-non-PH) and with pulmonary hypertension (COPD-PH).**

| Variable (unit) | Controls (n = 34) | COPD-non-PH (n = 74) | COPD-PH (n = 26) |
|---|---|---|---|
| **Global LV function** | | | |
| MPI septal (no unit) | 0.37±0.08 | 0.54±0.11* | 0.63±0.16* [†] |
| MPI lateral (no unit) | 0.36±0.06 | 0.56±0.12* | 0.60±0.10* [†] |
| MPI (no unit) | 0.36±0.05 | 0.55±0.10* | 0.62±0.10* [†] |
| Isovolumic contraction time (ms) | 59±11 | 76±12* | 72±13* [†] |
| Isovolumic relaxation time (ms) | 49±13 | 83±15* | 89±17* [†] |
| Ejection time (ms) | 299±20 | 291±33* | 265±31* [†] |
| Isovolumic relaxation time, HR adjusted | 54±15 | 95±23* | 115±25* [†] |
| **LV systolic function** | | | |
| Strain septal (%) | -21.9±2.2 | -15.8±2.0* | -13.8±1.3*[†] |
| Strain lateral (%) | -22.5±2.4 | -15.7±2.1* | -15.0±2.9* [†] |
| Strain (%) | -22.2±1.9 | -15.8±1.9* | -14.4±1.7* [†] |
| **LV diastolic function** | | | |
| Transmitral peak early (E) velocity (m/s) | 0.66±0.14 | 0.70±0.16 | 0.73±0.15 |
| Transmitral late (A) velocity (m/s) | 0.67±0.14 | 0.68±0.19 | 0.72±0.21 |
| Transmitral E/A ratio (no unit) | 1.07±0.29 | 1.04±0.25 | 1.03±0.26 |
| E´ (cm/s) | 8.9±2.3 | 8.3±2.0 | 7.4±1.7* |
| E/E´ ratio (no unit) | 7.7±1.5 | 8.6±2.1 | 10.3±2.2*[†] |
| **LV, left and right atrial dimensions** | | | |
| LV diastolic diameter (cm) | 5.1±0.6 | 4.8±0.7 | 4.5±0.6* |
| LV interventricular septum dimension (cm | 0.8±0.1 | 0.9±0.1 | 0.9±0.2 |
| LV posterior wall dimension (cm) | 0.8±0.1 | 0.9±0.1 | 0.9±0.1 |
| LV Mass (g/m$^2$) | 78±19 | 79±22 | 70±17 |
| Relative wall thickness (no unit) | 0.33±0.06 | 0.38±0.07* | 0.38±0.07* |
| Left atrium volume (ml/ m$^2$) | 21±4 | 24±5* | 21±4 [†] |
| Right atrium volume (ml/m$^2$) | 14±3 | 21±6* | 22±8* |
| Ratio of left/right atrium volumes | 1.6±0.5 | 1.3±0.5* | 1.1±0.3* [†] |

Values are mean±SD. E´: average of peak early mitral velocities from at the root of septal and lateral mitral leaflet, respectively, HR: heart rate, Isovolumic contraction and relaxation time and ejection time: average of measurements from the root of septal and lateral mitral leaflet, respectively, LV: left ventricular, MPI septal and MPI lateral: myocardial performance index measured at the root of septal and lateral mitral leaflet, respectively, MPI: average of MPI septal and lateral, strain septal and lateral: strain measured at basal third of LV septal and lateral wall, respectively, strain: average of strain septal and lateral, TDI: Tissue Doppler imaging.

* Significantly different from controls,

[†] significantly different from COPD-non-PH.

## LV systolic function

Both LV MPI and LV strain were significantly different between controls, COPD-non-PH and COPD-PH (p<0.01 for all) (Table 3 and Fig 1). LV MPI ≥0.51 was found in 64.9% and 88.5% and LV strain ≤-15.8 in 62.2.% and 76.9% in the COPD-non-PH and COPD-PH patients, respectively. When patients with diabetes and/or hypertension (n = 37) were excluded, this was not significantly changed. Even those with mPAP below 20 mmHg showed LV MPI and LV strain significantly impaired compared to controls (Table 4).

LV and RV ejection fraction by CMR correlated significantly, r = 0.66 (p<0.001), and there were significant correlations between LV MPI vs RV MPI and tricuspid annular plane systolic excursion of r = 0.70 and r = 0.59, respectively and between LV strain vs. RV MPI and tricuspid annular plane systolic excursion of r = 0.56 and r = 0.60 (p<0.001 for all), respectively. Fig

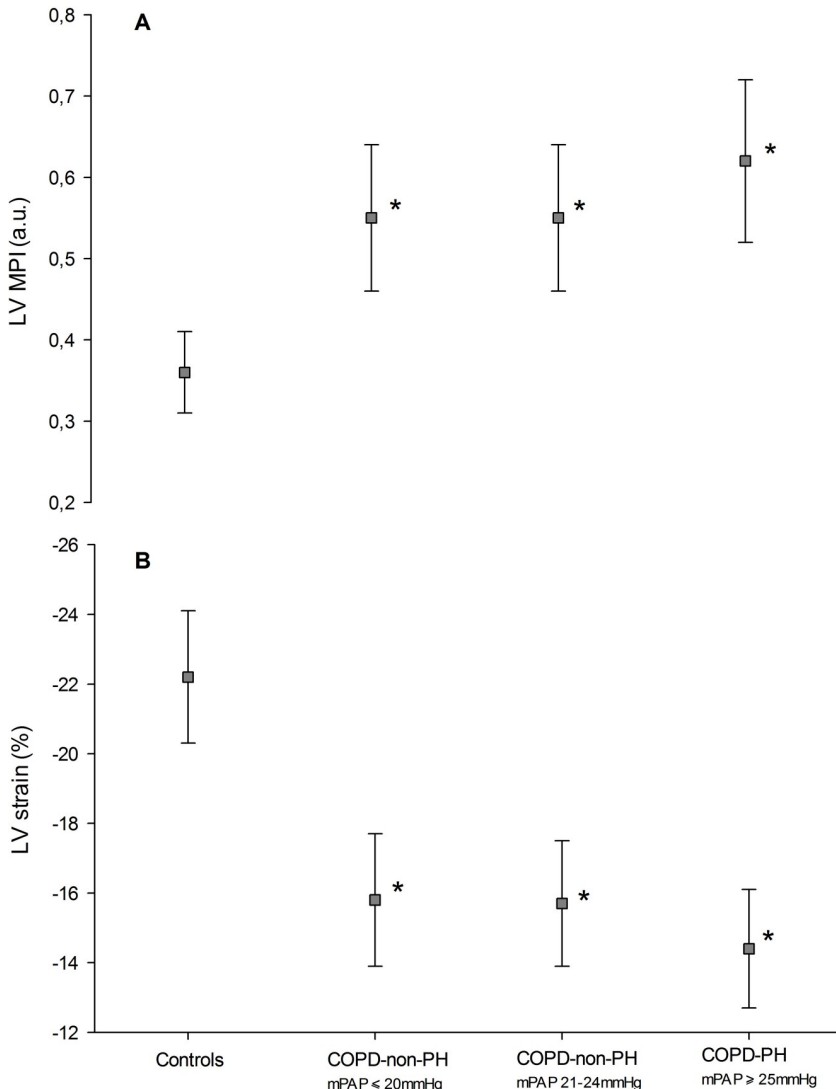

**Fig 1. LV myocardial performance index and LV strain in controls, COPD patients with normal and intermediate pulmonary pressure and with pulmonary hypertension.** Similar findings of LV dysfunction by LV myocardial performance index (MPI) (panel A) and LV strain (panel B) in 100 COPD patients divided in three groups; those with normal (≤20 mmHg) and borderline (21–24 mmHg) mean pulmonary pressure (mPAP) and those with pulmonary hypertension (≥25 mmHg) compared to 34 healthy controls. *significantly different from controls (p<0.01), †significantly different from COPD-non-PH with mPAP≤20 mmHg and mPAP 21–24 mmHg (both p<0.05).

2 displays significant correlations between stroke volume by right heart catheterization and residual volume % predicted, LV MPI and LV strain. There was no significant correlation between the LV/RV ratio by CMR and LV MPI and strain.

Two separate multiple regression models were fitted, and backward stepwise approach was used. In the final model with LV MPI as dependent variable, residual volume % predicted (p<0.01), invasive stroke volume (p<0.001), RV ejection fraction by CMR (p<0.01) and log transformed CRP (p<0.05) were all independently associated with LV MPI, and this model explained 55% of the variation in LV MPI. The initial model was also adjusted for systolic blood pressure, pack-years, heart rate, interleukin 6, forced expiratory volume in the first second, forced expiratory volume in the first second/forced vital capacity, arterial oxygen tension

**Table 4. LV function by LV MPI and LV strain, key respiratory and invasive hemodynamic indices of COPD patients with no pulmonary hypertension divided in two groups, mean pulmonary pressure ≤20 mmHg (normal) and 21–24 mmHg (borderline).**

| | Controls (n = 34) | COPD (n = 53) | COPD (n = 21) |
|---|---|---|---|
| **Global (MPI) and LV systolic function** | (n = 34) | mPAP≤20 mmHg | mPAP 21–24 |
| MPI septal (no unit) | 0.36±0.08 | 0.55±0.12* | 0.52±0.10* |
| MPI lateral (no unit) | 0.37±0.06 | 0.56±0.12* | 0.58±0.11* |
| MPI (no unit) | 0.36±0.06 | 0.55±0.09* | 0.55±0.08* |
| Strain septal (%) | -21.9±2.2 | -15.8±2.1* | -15.7±1.7* |
| Strain lateral (%) | -22.5±2.4 | -15.7±2.1* | -15.7±2.1* |
| Strain (%) | -22.2±1.9 | -15.8±1.9* | -15.7±1.8* |
| **Hemodynamic/respiratory variables** | | | |
| Mean pulmonary pressure (mmHg) [z] | | 16.7±2.5 | 22.6±1.2[†] |
| Pulmonary vascular resistance (Wu) [z] | | 1.8±0.7 | 2.4±1.0[†] |
| Cardiac output (l/min) [z] | | 5.1±1.0 | 5.2±0.9 |
| Pulmonary artery wedge pressure (mmHg) [z] | | 7.7±3.5 | 9.9±3.5 [†] |
| PaO$_2$ (kPa) | | 10.0±1.3 | 9.5±1.0 |
| Residual volume% predicted | 119±16 | 182.5±52.2* | 212.9±69.1[†] |
| FEV$_1$% predicted | 98±10 | 50.8±15.6* | 38.1±14.1[†] |

PaO$_2$: Arterial oxygen tension, FEV$_1$: Forced expiratory volume in the first second, MPI: average of MPI septal and lateral, Strain: average of strain septal and lateral,

[*] Significantly different from controls ($p<0.01$),

[†] Significantly different from COPD with mPAP≤20 mmHg ($p<0.05$) [z] Performed by right heart catheterization.

and mPAP, which were all statistically significant ($p<0.01$ for all) in univariate analyses. Similarly, with LV strain as dependent variable: Residual volume % predicted ($p<0.01$), invasive stroke volume ($p<0.001$), mPAP ($p<0.05$), and forced expiratory volume in the first second/forced vital capacity ($p<0.05$), explained 57% of the variation in LV strain. In addition to the above first six listed independent variables, the initial model was also adjusted for RV ejection fraction, arterial oxygen tension and log transformed CRP. All nine variables, except for heart rate and arterial oxygen tension, were statistically significantly associated with LV strain in univariate analyses ($p<0.05$ for all). In addition, we included sex and age in both models (both not statistically significant in crude analyses).

### LV diastolic function

There was a small, but statistically significant increase of E/E´ from controls to COPD-non-PH ($p<0.01$), and to COPD-PH ($p<0.01$), driven by the reduced E´. The E/A ratio and left atrial volume did not differ between the three groups (Table 3). However, isovolumic relaxation time by TDI, without and with adjustment for heart rate, was significantly increased in both patient groups compared to controls and between COPD-PH and COPD-non-PH (Table 3). Isovolumic relaxation time ≥87ms was found in 42% and 58% of the COPD-non-PH and COPD-PH patients, respectively. When adjusted for heart rate <99, this was changed to 36% and 64%, respectively. There were significant, but weak correlations between invasive stroke volume and heart rate adjusted isovolumic relaxation time, r = - 0.46 ($p<0.01$), left atrial volume, r = 0.32 ($p<0.01$) and the E/A ratio, r = 0.21 ($p<0.05$), however not to E, E´ and E/E´.

### LV and RV volumes

Table 5 shows LV and RV volumes and ejection fraction by CMR, two- and three dimensional echo. End diastolic LV/RV volume ratio was significantly lower, driven by RV volume, in both

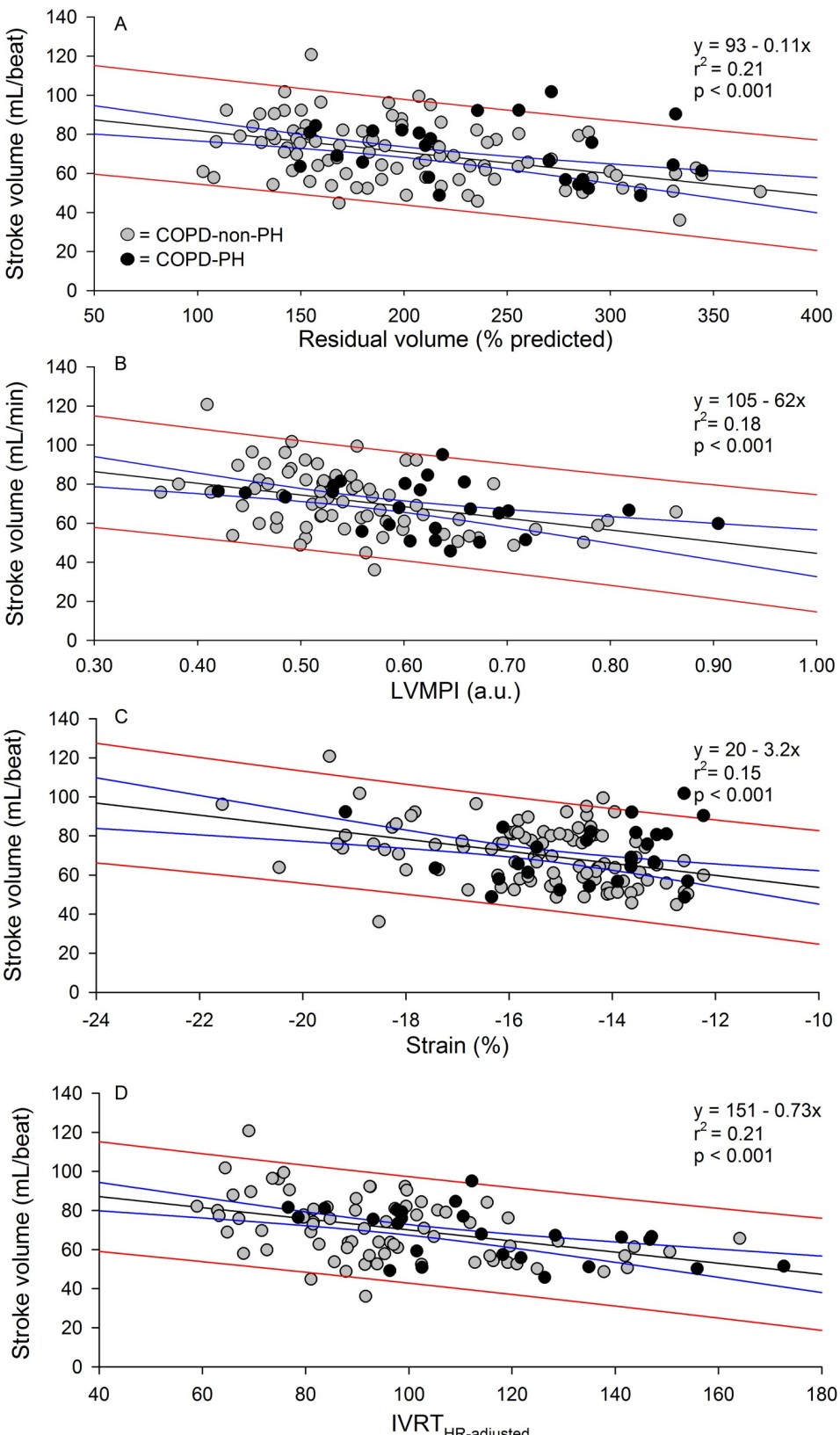

**Fig 2. Associations between stroke volume and residual volume, LV myocardial performance index, LV strain and heart rate adjusted isovolumic relaxation time.** Panel A, B, C and D display significant associations between invasive stroke volume and residual volume (% predicted), LV MPI (myocardial performance index), LV strain and IVRT HR-adjusted (heart rate adjusted isovolumic relaxation time), respectively. Grey circles are COPD patients without pulmonary hypertension (COPD-non-PH) and black circles COPD patients with pulmonary hypertension (COPD-PH).

COPD-groups compared to controls; controls: 1.03, COPD-non-PH: 0.88, and COPD-PH: 0.78 (p<0.01 for all). Similar to the results with three-dimensional echo, the LV/RV ratio of end diastolic volumes by CMR decreased from 0.88 in COPD-non-PH to 0.81 in COPD-PH (p<0.01). LV ejection fraction by CMR <50% was found in 19%, although mild, 45±5%.

## Discussion

The present study has demonstrated subclinical impairment of LV global and systolic function by TDI in a stable cohort of COPD patients without cardiovascular disease compared to

**Table 5. Left and right ventricle volumes and ejections fractions in healthy controls, patients with chronic obstructive pulmonary disease without (COPD-non-PH) and with (COPD-PH) pulmonary hypertension.**

| Variables (unit) | Controls (n = 34) | COPD-non-PH (n = 68) | COPD-PH (n = 20) |
|---|---|---|---|
| **Left ventricle CMR** | | | |
| End-diastolic volume (ml/m²) | | 75±15 | 70±12 |
| End-systolic volume (ml/m²) | | 34±12 | 35±13 |
| Stroke volume index (ml/beat/m²) | | 42±9 | 39±9 |
| Myocardial Mass (g/m²) | | 41±15 | 43±21 |
| Ejection fraction (%) | | 56±7 | 56±10 |
| **Right ventricle CMR** | | | |
| E End-diastolic volume (ml/m²) | | 88±26 | 95±25 |
| End-systolic volume (ml/m²) | | 45±14 | 52±16 |
| Stroke volume index (ml/beat/m²) | | 43±14 | 43±13 |
| Ejection fraction (%) | | 49±6 | 45±9 |
| **Left ventricle 3DE** | | COPD-non-PH (n = 74) | COPD-PH (n = 26) |
| End-diastolic volume (ml/m²) | 59±10 | 61±12 | 57±11 |
| End-systolic volume (ml/m²) | 23±5 | 26±7* | 25±7 |
| Stroke volume index (ml/beat/m²) | 36±6 | 35±7 | 33±7 |
| Ejection fraction (%) | 62±5 | 58±5* | 58±5* |
| **Right ventricle 3DE** | | | |
| End-diastolic volume (ml/m²) | 57±7 | 71±15* | 74±15* |
| End-systolic volume (ml/m²) | 24±5 | 35±9* | 40±11* |
| Stroke volume index (ml/beat/m²) | 34±4 | 35±7 | 34±7 |
| Ejection fraction (%) | 58±4 | 50±5* | 46±6*ʸ |
| **Left ventricle 2DE** | | | |
| End-diastolic volume (ml/m²) | 55±9 | 54±11 | 50±12 |
| End-systolic volume (ml/m²) | 22±5 | 23±6 | 22±7 |
| Stroke volume index (ml/beat/m²) | 33±5 | 31±6 | 29±6* |
| Ejection fraction (%) | 61±5 | 57±4* | 58±5* |

Values are mean±SD. CMR: Magnetic resonance imaging, 3DE and 2D: Three and two dimensional echocardiography, respectively,

*significantly different from controls,

ʸsignificantly different from COPD-non-PH.

controls. These findings were present in COPD patients with and without pulmonary hypertension, even in those with mPAP below 20 mmHg.

## LV systolic function

LV systolic and global function, however, as measured by TDI strain and LV MPI, respectively, were both able to differentiate between controls and the COPD subgroups. Moreover, the impairment of LV function by these two echo indices was similar on the LV lateral wall as that of the interventricular septum. The results also demonstrated that LV systolic dysfunction was highly prevalent, as the majority of our patients demonstrated LV MPI and strain above and below the predefined cut-off of ≥0.51 and <-15.8%, respectively. The proportion of patients with LV dysfunction remained high, even when those with cardiovascular risk factors (diabetes and hypertension) were excluded. Moreover, traditional method as 2D and novel 3D ejection fraction were also reduced in COPD with and without PH as compared to normal individuals. The difference, however, was small and probably not of clinical significance, but emphasizes the findings of LV strain and MPI. In addition, LV ejection fraction by CMR was also found mildly reduced in 19%.

Freixa et al. showed a prevalence of LV dysfunction of 13.3%, and Macchia et al. 13.8% [19, 20]. In the study by Frexia et al., however, there was more than 60% self-reported LV disease, and in the study by Macchia et al., patients with LV disease were not excluded. These two studies reflect the main difference between the present study and other studies on COPD, namely the thorough exclusion of clinical LV disease, the absence of LV ejection fraction <50% prior to inclusion, the use of more sensitive TDI echo tools and invasive hemodynamic data in the present study. We consider this to emphasize that our findings most likely are caused by the COPD disease itself and not by other left heart diseases. Although a recent study reported similar prevalence of coronary artery disease in COPD as in an age and gender matched group without pulmonary disease, we cannot exclude that some of our COPD patients had silent coronary heart disease [21].

Several studies have shown that the main reason for the reduction of LV function in advanced COPD patients is the right side of the heart due to impaired RV function, dysfunctional interventricular septum due to increased pulmonary pressure, oversized right heart, and thus under-filling of the left ventricle [1–4, 6, 22]. The present multiple regression analyses confirm the connection between impaired LV function by the two novel indices and the hyperinflated lungs, reduced stroke volume, and impaired RV function, and that this underscores the notion of under-filling as the main mechanism for the subclinical reduction in LV function.

The present study has demonstrated that even those with mPAP below 20 mmHg, showed increased LV MPI and reduced strain, and that the LV systolic impairment by these two indices was similar between the LV interventricular septum and the LV lateral wall. We consider this to reflect that the abovementioned systemic under-filling has a more negative and systemic impact on LV function than that of the interventricular septum. The not existing association between the LV/RV ratio by CMR and the two TDI indices for LV function underscores this notion. Nevertheless, our results of early impairment of LV systolic function in COPD are in line and extend the large and non-invasive study by Barr et al. showing a gradual impairment of LV systolic function from normal lung structure to severe air flow obstruction [4].

It has been speculated, since there is substantial loss of pulmonary vascular bed even in smokers and mild COPD, that this may lead to a small increase in pulmonary vascular resistance and reduced compliance leading to an increase of RV load and reduction of RV function and thus under-filling of the left side [3, 4, 23]. It has also been suggested that the systemic

inflammatory state in patients with COPD may cause LV functional changes [24], and that smoking may have a negative direct impact on LV function [25]. However, we could not show any reliable association to inflammation parameters or to pack-years, respectively.

Our patients had a low-grade state of inflammation by higher circulating levels of inflammatory biomarkers compared to healthy controls. There were, however, no differences between COPD-non-PH and COPD-PH, which confirms COPD as a pro inflammatory disease [24, 26]. However, only crp remained as an independent variable in the multiple regression analyses to predict LV function, in contrast to a previous study, showing that IL-6 was higher in COPD-PH than COPD-non-PH, and a correlation with severity of pulmonary hypertension [26].

### LV diastolic function

The main echo indices for the diagnosis of LV diastolic dysfunction are the size of the left atrium, E´, E/A, and E/E´ ratios [17]. Our COPD-PH patients showed higher E/E´ ratio and lower E´; however, only on a group level. In particular, only eight patients had E/E´ above 13, and none had a pulmonary wedge pressure above 15 mmHg by right heart catheterization or a left atrium > 34 ml/ m² [17]. Although we could show a marked increase of isovolumic relaxation time in our patients (Table 3), indicating impairment of early diastolic LV relaxation, there was not a consistent link between LV diastolic dysfunction and COPD based on standard echo indices. Moreover, left atrial and LV size were functionally undersized compared to the right side, which probably, in concert with the impaired RV function and the significant associations between stroke volume and left atrium volume, reflects under-filling of the left side (Table 3). Although under-filling probably is not the only mechanism for the reduced size of the left side, this may have masked LV diastolic dysfunction as assessed by standard echo indices in several of our patients. To the best of our knowledge, we could not find any study showing LV diastolic dysfunction in the majority of the COPD patients as measured by standard echo indices, only small group differences [6, 10, 27]. This in contrast to what is expected in this group of patients [24]. LV isovolumic relaxation time by TDI, in particular heart rate adjusted, seems to be the best method to evaluate LV diastolic dysfunction in patients with stable COPD disease, which is consistent with two other studies [6, 10]. Its association to pulmonary pressure is also important for the mechanistic understanding of this echo index.

### Limitations

The study participants were not routinely screened invasively with coronary angiograms, and silent coronary artery disease could therefore be overlooked. Although a thorough screening process and testing with echocardiography, resting ECG and exercise testing were performed to unmask ischemic heart disease, we cannot entirely exclude neither macro nor microvascular disease in our patients.

We included medically treated hypertension, diabetes and smokers, known risk factors for cardiovascular disease, and hence, confounders in the evaluation of LV function and structure. However, restricting the analyses to participants with or without systemic hypertension, and to the presence or absence of diabetes, gave similar results.

We did not perform CMR in the controls. The absence of CMR data in this group limits a definite assessment of the role of COPD in the pathogenesis of cardiac disorders in relation to hyperinflation and LV mass.

The present study reports upper normal limits for LV MPI, LV Strain, isovolumic relaxation time and isovolumic relaxation time adjusted for heart rate. Determining accurate thresholds from continuous variable have many pitfalls as pointed out by Giannoni m and co-

workers [28]. Although we made three times standard deviation of the mean from the controls to avoid false positive results, it does not necessarily provide our study with the correct upper normal limits.

## Conclusion

LV systolic dysfunction by LV MPI and strain, was highly prevalent even in COPD patients without pulmonary hypertension. LV diastolic dysfunction when measured with conventional methods, was hardly present. Isovolumic relaxation time by TDI, however, seems to be a consistent echo index to reveal LV diastolic dysfunction in this stable cohort of COPD.

## Supporting information

**S1 Datafile.**
(DAT)

## Author Contributions

**Conceptualization:** Janne M. Hilde, Viggo Hansteen, Sjur Humerfelt, Kjetil Steine.

**Data curation:** Janne M. Hilde, Jonny Hisdal, Ingunn Skjørten, Viggo Hansteen, Morten N. Melsom, Ole J. Grøtta, Milada C. Småstuen, Sjur Humerfelt, Kjetil Steine.

**Formal analysis:** Janne M. Hilde, Ole J. Grøtta, Ingebjørg Seljeflot, Harald Arnesen, Kjetil Steine.

**Investigation:** Janne M. Hilde, Ingunn Skjørten, Morten N. Melsom, Sjur Humerfelt.

**Methodology:** Janne M. Hilde, Jonny Hisdal, Ingunn Skjørten, Morten N. Melsom, Sjur Humerfelt, Kjetil Steine.

**Resources:** Jonny Hisdal.

**Supervision:** Viggo Hansteen, Ingebjørg Seljeflot, Harald Arnesen, Sjur Humerfelt, Kjetil Steine.

**Validation:** Ingebjørg Seljeflot, Harald Arnesen, Kjetil Steine.

**Visualization:** Jonny Hisdal.

**Writing – original draft:** Janne M. Hilde.

**Writing – review & editing:** Kjetil Steine.

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
