## [Decision Letter · Decision Letter 0]

31 Mar 2020

PONE-D-20-04365

Left ventricular dysfunction in COPD without pulmonary hypertension

PLOS ONE

Dear Ass. Professor Steine,

Thank you for submitting your manuscript to PLOS ONE. After careful consideration, we feel that it has merit but does not fully meet PLOS ONE’s publication criteria as it currently stands. Therefore, we invite you to submit a revised version of the manuscript that addresses the points raised during the review process.

ACADEMIC EDITOR: All issues raised by expert reviewers are required.

We would appreciate receiving your revised manuscript by May 15 2020 11:59PM. To enhance the reproducibility of your results, we recommend that if applicable you deposit your laboratory protocols in protocols.io, where a protocol can be assigned its own identifier (DOI) such that it can be cited independently in the future. For instructions see: http://journals.plos.org/plosone/s/submission-guidelines#loc-laboratory-protocols

We look forward to receiving your revised manuscript.

Kind regards,

Vincenzo Lionetti, M.D., PhD

Academic Editor

PLOS ONE

Journal Requirements:

2. Please note that according to our submission guidelines (http://journals.plos.org/plosone/s/submission-guidelines), outmoded terms and potentially stigmatizing labels should be changed to more current, acceptable terminology. For example: “Caucasian” should be changed to “white” or “of [Western] European descent” (as appropriate).

Reviewers' comments:

Reviewer's Responses to Questions

**Comments to the Author**

1. Is the manuscript technically sound, and do the data support the conclusions?

Reviewer #1: Yes

Reviewer #2: Yes

2. Has the statistical analysis been performed appropriately and rigorously? 

Reviewer #1: Yes

Reviewer #2: Yes

3. Have the authors made all data underlying the findings in their manuscript fully available?

Reviewer #1: Yes

Reviewer #2: Yes

4. Is the manuscript presented in an intelligible fashion and written in standard English?

Reviewer #1: Yes

Reviewer #2: Yes

5. Review Comments to the Author

Reviewer #1: In the paper of Steine and coworkers the prevalence of left ventricular (LV) systolic and diastolic

function in a stable cohort of COPD patients (n=100) with and without pulmonary hypertension (PH) was thoroughly evaluated by using both advanced echocardiography and cardiac MRI. Authors found that subclinical LV systolic dysfunction, as expressed by altered LV myocardial performance index (MPI ≥0.51) and LV strain (≤15.8%.) was a frequent finding in this cohort of COPD patients, even in those with normal pulmonary artery pressure. On the contrary, LV diastolic dysfunction was hardly documented in COPD patients.

The authors should be commended for the great effort in collecting several instrumental (including MRI and right heart catheterization) and biohumoral data and for the solid methodological approach used in a relatively large cohort of patients. However, I have the following concerns:

1) The hypothesis behind the study seems week. The reason why it would be clinically important to screen with that attention to details COPD patients to identify signs of subclinical myocardial damage should be better highlighted and explained. This is of key importance to understand the clinical repercussions of the study. Authors conclude “LV MPI and strain are two simple echo indices of LV global and systolic function, respectively, that should be implemented in the examination of patients with COPD. Since most COPD patients have subclinical LV dysfunction, this should be taken into consideration when the COPD patients´ dyspnea is considered by the clinicians.” However, the symptoms or the functional capacity (I mean 6MWT or better a cardiopulmonary exercise testing) have not been performed and thus it is unknown whether the decrease in MPI or LV strain at rest would actually contribute (and to which extent) to the symptoms and/or outcome of COPD patients.

2) A resting ECG or a dynamic exercise test have poor sensitivity to exclude comorbid coronary artery disease. This may be a major limitation of the study also considering that the great majority of patients were smokers or ex-smokers with a relevant exposition to smoking. Therefore, it is likely that a high percentage of patients do have macrovascular or microvascular coronary artery disease and thus the subclinical LV systolic dysfunction has to be related to smoking or CAD (not recognized by a simple ECG stress testing).

3) Authors should justify the choice of using TDI strain analysis only on the basal third of the septal and lateral LV walls, instead of considering the whole ventricle. What if some apical spearing was present? Considering a tomtec workstation why do not use speckle-tracking on the whole ventricle?

4) Please discuss also the potential importance of left atrial strain especially considering the strong association with symptoms (Cameli Int J Cardiol. 2019;286:87-91)

5) Surprisingly, no data on late gadolinium enhancement or T1-mapping is provided relative to cardiac MRI analysis. To use only cardiac MRI to obtain LV ejection fraction or LV RV ratio is a serious underuse of a much powerful technique. Is there any macro or microscopic fibrosis at MRI. Please report at least the LGE data.

6) Even more surprisingly authors have not assessed two biomarkers that can easily disclose a subclinical myocardial involvement, even more sensitive than echo and cardiac MRI, i.e. natriuretic peptides and high sensitivity troponins. This is a major limitation and seems not logical in such a complex study design including cardiac MRI and right heart catheterization. Do you have those data?

7) Again in the results the order and meaning of the several correlations provided seems mainly explorative and not supported by an overall hypothesis.

8) Please avoid comments in the results and leave them to the discussion (i.e. which explains the significant difference in cardiac index”).

9) Patients using beta-blokers, Warfarin or clopidogrel were also excluded. What about aspirin (are there any patients with CAD)? What about other drugs used in hypertension such as ACE-inhibitors, ARBs, MRA that can have an impact on myocardial performance?

10) Please show a CONSORT diagram to make the reader (and reviewers) aware of any potential selection bias. In a population of smokers as the one presented is very unusual to not find some atherosclerotic disease, either on carotid artery or coronary artery and so on. How many patients with CAD were excluded from the original population?

11) Finally, please specify the way the cutpoint of ≥0.51and <-15.8% were “pre”-defined, since this is of key relevance in defining the epidemiology of subtle LV dysfunction in your population. Please be aware and discuss the need to use cutpoints defined in similar population of the one under study and the problem of losing power or clinical information when dicothomizing a continous variable (please refer to Giannoni, PLoS One 2014;9:e81699.).

12) In the discussion, I do not understand why authors attribute the loss in MPI and LV strain to underfilling also in patients with completely normal pulmonary artery pressure. Why should those patients have a problem of the underfilling? Are there actually any significant differences in pulmonary vascular resistances in your population? From the data you showed patients with COPD and normal pulmonary pressure have slightly higher LV systolic volumes as compared to controls (underfilling?) Is there any relationship between PVR and MPI or LV strain? Please avoid overstatement and keep the discussion closer to the data you have. The problem of inflammation in this respect seems more promising and supported by data.

13) Please give more relevance also in the results and the discussion to the decrease observed in 2D and 3D LVEF and RVEF, do not focus only on MPI and LV strain, this seems a biased way of presenting your data.

Reviewer #2: I have read with great interest the article regarding LV dysfunction in COPD patients without pulmonary hypertension. The novelty in research paper is slightly dubious. Researches have already studied this problem. For example- Willem-Jan Flu A et al (Co-existence of COPD and left ventricular dysfunction in vascular surgery patients. Respiratory Medicine (2010) 104, 690-696)

have shown that mild COPD and subclinical LV dysfunction often coexist, the combination of these two is associated with an increased risk for long-term all-cause mortality. Also authors have mentioned about using preoperative ‘integrated cardiopulmonary risk index’ by performing standard preoperative spirometry and echocardiography.

Current study confirms abovementioned findings about coexisting of COPD and subclinical LV dysfunction, but also adds mechanisms for the reduced LV systolic function in COPD patients without pulmonary hypertension. Researchers have found that LV MPI and strain are two simple echo indices of LV global and systolic function in these patients and based on this they recommend to implement it in the examination of patients with COPD. In the light of the above; this study may be helpful in COPD management and can be used as simplified COPD management tool.

6. PLOS authors have the option to publish the peer review history of their article (what does this mean?). If published, this will include your full peer review and any attached files.

Reviewer #1: No

Reviewer #2: No

---

## [Author Response · Author response to Decision Letter 0]

14 May 2020

Answers to Reviewer #1 concerning the article

Left ventricular dysfunction in COPD without pulmonary hypertension 

1) The hypothesis behind the study seems week. The reason why it would be clinically important to screen with that attention to details COPD patients to identify signs of subclinical myocardial damage should be better highlighted and explained. This is of key importance to understand the clinical repercussions of the study. Authors conclude “LV MPI and strain are two simple echo indices of LV global and systolic function, respectively, that should be implemented in the examination of patients with COPD. Since most COPD patients have subclinical LV dysfunction, this should be taken into consideration when the COPD patients´ dyspnea is considered by the clinicians.” However, the symptoms or the functional capacity (I mean 6MWT or better a cardiopulmonary exercise testing) have not been performed and thus it is unknown whether the decrease in MPI or LV strain at rest would actually contribute (and to which extent) to the symptoms and/or outcome of COPD patients

Answer: 

Hypothesis: We agree with the comment from the reviewer that the hypothesis should be more highlighted and explained more thoroughly, in particular why it is important to discover subclinical LV systolic dysfunction. To overcome this request, we had to change the introduction substantially by adding new sentences and making deletions not only to be within similar number of words as the first edition, but also in the sense of highlighting the novel content better. The reviewer can in that regard study page four and five in the attached and revised article, where the novel sentences are marked in red and deletions will be found.

 We agree with the reviewer that 6MWT and in particular cardiopulmonary exercise testing (CPET) are good functional tests in COPD patients and had fit in the article. Both tests have been performed as parts of a larger study including two PhDs and eight articles. These results are, however, not implemented in the present study partly because it has been presented before (1) and partly because the overall scope of this article had become too large. 

2) A resting ECG or a dynamic exercise test have poor sensitivity to exclude comorbid coronary artery disease. This may be a major limitation of the study also considering that the great majority of patients were smokers or ex-smokers with a relevant exposition to smoking. Therefore, it is likely that a high percentage of patients do have macrovascular or microvascular coronary artery disease and thus the subclinical LV systolic dysfunction has to be related to smoking or CAD (not recognized by a simple ECG stress testing).

Answer: We thank you for this comment. Cazolla M. et al found in a population-based retrospective study that COPD had 14% increased risk for ischemic events vs. 7% in the general population (2). On the other hand, Hong Y and co-workers studied more than 26.000 coronary angiograms retrospectively and found that odds ratio of having CAD was 0.83 for patients with COPD compared to those without COPD (3). The sensitivity of exercise testing ranges between 60 and 70%, while specificity has been reported between 85 and 90%(4). In addition to the regular dynamic exercise test, all our patients went through a cardiopulmonary exercise test (1), and all who could not perform a bicycle exercise test or perform a six minutes walking test, were excluded from the study. Although we cannot exclude that some of our patients had macrovascular coronary disease, and that these patients may have had a negative influence on LV systolic function, we do not believe this to have any important impact on our finding of subclinical LV dysfunction. We have, however, rephrased the second sentence in the Limitation subchapter, see below and on page 14, second paragraph: 

 When it comes to microvascular coronary artery disease, we do agree that this might be a possible and theoretical explanation for the subclinical LV dysfunction in COPD. We could not, however, find any study that has demonstrated such a direct connection in COPD. One study, however could show increased retinopathy in COPD (5). More interesting is the HFPEF paradigm introduced by Paulus WJ and Tschope C, who claim that patients with overweight/obesity, diabetes mellitus, chronic obstructive pulmonary disease, and hypertension induce a systemic proinflammatory state that causes coronary microvascular endothelial inflammation 

which reduces nitric oxide bioavailability, cyclic guanosine monophosphate content, and in the end protein kinase G (PKG) activity in cardiomyocytes favoring hypertrophy development and increases resting tension (6) It is tempting to argue that such a mechanism also could be responsible for LV systolic subclinical dysfunction as presented in the present study. On the other hand, our COPD patient could hardly show any LV diastolic dysfunction that often precedes HFPEF.

Page 14, second paragraph: Although a thorough screening process and clinically relevant non-invasive testing with echocardiography, resting ECG and exercise testing were performed to unmask ischemic heart disease, we cannot entirely exclude neither macro nor microvascular disease in our patients. 

3) Authors should justify the choice of using TDI strain analysis only on the basal third of the septal and lateral LV walls, instead of considering the whole ventricle. What if some apical spearing was present? Considering a tomtec workstation why do not use speckle-tracking on the whole ventricle?

Answer: We chose LV basal TDI strain of two reasons: The TDI strain findings of the basal part of the left ventricle is a summation of the systolic function or strain of the whole left ventricle. If the TDI strain is reduced further down towards the apex in the left ventricle, the basal strain or systolic deformation would also be reduced. A similar principle is used for systolic LV basal velocity and mitral annular plane systolic excursion (MAPSE) with M-mode

 The second reason is a consequence of only performing strain measurement at the LV base: It is for the sake of simplicity, because we considered it for important to have reliable measurements that were easy to use. If apical spearing had been present as by amyloidosis, the LV basal strain would have been reduced as well. However, such patients would have been excluded in any case by the echo examination before inclusion. 

 Two-dimensional speckle tracing was not invented when this study was performed. In this respect we want to underline the following: It is well known that patients with COPD in most cases have reduced acoustic conditions. When two-dimensional speckle tracking is used, there will be a problem to achieve two-dimensional loops of good enough quality and thus impossible to measure strain on quite few of these patients. In the the ACE 1950 study, a community study including 3706 individuals of both sexes in their mid-sixties, we could measure LV two-dimensional strain in only 67% of the participants (7) 

4) Please discuss also the potential importance of left atrial strain especially considering the strong association with symptoms (Cameli Int J Cardiol. 2019;286:87-91)

Answer: The study by Cameli M and co-workers have demonstrated a strong and negative correlation between quality of life (The Minnesota Living with Heart Failure Questionaire (MLHFQ)) and left atrial reservoir strain (8). The physical basis for this connection is that atrial stretch receptors are stimulated by increased filling pressure in patients with heart failure. This information is then translated by vagal afferent nerves to medulla and the limbic system where the perception of dyspnoea occurs. As mentioned in this paper, the size of the left atrium is well established as a prognostic factor, the larger size the worse prognosis. The paper by Cameli M et al. have thus given us a mechanism or connection (impaired reservoir atrial function) between experienced dyspnoea and quality of life in patients with heart failure. 

 It is tempting to see the paper by Cameli M et al in light of the present study, since dyspnoea is a main symptom in COPD. As mentioned on page 13, paragraph four, we could not show any clear evidences, neither by echo nor by right heart catheterisation (pulmonary wedge pressure) that our COPD patients had increased LV filling pressure. Although it should have been very interesting to perform left atrial reservoir strain on our patients to find if their dyspnoea partly is related to left atrial strain (or right atrium), it is to be expected that dyspnoea in COPD mainly is caused by the lungs.

5) Surprisingly, no data on late gadolinium enhancement or T1-mapping is provided relative to cardiac MRI analysis. To use only cardiac MRI to obtain LV ejection fraction or LV RV ratio is a serious underuse of a much powerful technique. Is there any macro or microscopic fibrosis at MRI. Please report at least the LGE data.

Answer: We agree that late gadolinium enhancement (LGE) data and measurements of fibrosis had been interesting in this respect, in particular in view of possible mechanism for the subclinical LV dysfunction. The MRI examinations were performed 11 years ago, and the protocol was then set up merely for evaluation of volumes. No contrast was administered, and hence no LGE data are available. T1-mapping, as it is understood today, was not available when our original recordings were performed.

6) Even more surprisingly authors have not assessed two biomarkers that can easily disclose a subclinical myocardial involvement, even more sensitive than echo and cardiac MRI, i.e. natriuretic peptides and high sensitivity troponins. This is a major limitation and seems not logical in such a complex study design including cardiac MRI and right heart catheterization. Do you have those data?

Answer: We have measured natriuretic peptides. They were originally within our paper in JACC from 2013, but were taken out according to space problems in that article, and because they were within normal references (9). NT pro-BNP were 9.8 (9.1,10.5) pmol/l in those without pulmonary hypertension and 10.0 (9.1,10.8) in those with pulmonary hypertension. NT pro-BNP was thus normal in both groups, and there was not any clinically important correlation to LV dysfunction. This is now added in table 1 on page 20. Since the present article is about LV dysfunction, NT pro-BNP is the most relevant of the two. As high sensitive troponin (Hs tnt) is more interesting concerning prognostic implication, it is our plan to perform the analysis based on biobank samples by use of high sensitive assay. This will, however, be quite another study.

7) Again in the results the order and meaning of the several correlations provided seems mainly explorative and not supported by an overall hypothesis.

Answer: We partly agree to this, and it has been discussed thoroughly among the authors. Our conclusion is that these correlations are connected and belongs to the context of the article. We agree, however that they, rated one by one, not necessarily support the overall hypothesis, but they explain associations and possible mechanisms which again are important for the final result, namely the substantial subclinical systolic LV dysfunction in patients with COPD.

8) Please avoid comments in the results and leave them to the discussion (i.e. which explains the significant difference in cardiac index”).

Answer: We have done as recommended, and taken out the sentence as pointed out by the reviewer. See page nine, first paragraph. 

9) Patients using beta-blokers, Warfarin or clopidogrel were also excluded. What about aspirin (are there any patients with CAD)? What about other drugs used in hypertension such as ACE-inhibitors, ARBs, MRA that can have an impact on myocardial performance?

Answer: Patients on aspirin were only excluded if this were due to coronary artery disease, and they were, of course, thoroughly interviewed and examined with regards to CAD. As written on page six, first paragraph: ”Patients with history of congenital, rheumatic, valvular and ischemic heart disease, treated arterial hypertension with blood pressure >160/90 mmHg,….were also excluded.” This means that patients with well controlled hypertension who were on ACE-inhibitors (16 participants), ARBs (0 participants.) or MRA (0 participants.) were included. We do not consider these drugs to have negative impact on LV systolic function. Rather the other way around, since they are used in the treatment of LV dysfunction. If systemic blood pressure is not good enough controlled, this may have a negative impact on LV function. Moreover, excluding 37 participants with diabetes and/or hypertension did not change the prevalence of LV systolic dysfunction by LV MPI and LV strain (See page nine, second paragraph) As can be seen from table 1, a major part of the COPD patients were normotensive. Regarding patients with CAD, see point 2 above.

10) Please show a CONSORT diagram to make the reader (and reviewers) aware of any potential selection bias. In a population of smokers as the one presented is very unusual to not find some atherosclerotic disease, either on carotid artery or coronary artery and so on. How many patients with CAD were excluded from the original population?

Answer: As the overall intention with the present study was to find out if the COPD disease itself had a negative impact on LV function, it was important to exclude any other cause than COPD that could damage LV function. These potential selection bias have now been outlined in point 2) with the given changes in the manuscript page 14, second paragraph, and point 9).

In addition, there was in advance performed a thorough screening process, where approximately 1300 patients with COPD were screened by one of the authors (Ingunn Skjorten) very thoroughly. Those with the slightest symptoms of CAD, were excluded in this part of the process. Unfortunately, we do not have any exact number how many that were excluded according to this reason. We have therefore not included a CONSORT diagram. Those participants who came through this needle`s eye were then exposed to exercise bicycle test. None of these patients experienced horizontal or downwards sloping of the ST segment ≥ 1 mm. However, four got chest pain, of whom two had normal CT angiograms, one normal coronary angiogram and one had a borderline stenosis which was not dilated. 

11) Finally, please specify the way the cutpoint of ≥0.51and <-15.8% were “pre”-defined, since this is of key relevance in defining the epidemiology of subtle LV dysfunction in your population. Please be aware and discuss the need to use cutpoints defined in similar population of the one under study and the problem of losing power or clinical information when dicothomizing a continous variable (please refer to Giannoni, PLoS One 2014;9:e81699.).

Answer: We are thankful for making us aware of the problems of making threshold for a given continuous variable. As we have written in the method section on page seven, second paragraph: “mean+three standard deviations of the controls for all four were considered abnormal” were considered as the threshold or upper normal limit in our population. Although we considered by this to be well within the limit for not generating false positive results, since mean + 2SD is mostly used, we acknowledge, after studying the paper by Giannoni A and co-workers, that even then it is difficult to achieve reliable thresholds with continuous variables, which are highly depended on the mean of the different studies (9). According to this recognition, we have added Gianonni A et al as reference no 28 in the article and underlined this in the limitation section on page 15, paragraph three in the Limitation section by adding: 

Page 15, paragraph three: “The present study reports upper normal limits for LV MPI, LV Strain, isovolumic relaxation time and isovolumic relaxation time adjusted for heart rate. Determining thresholds from continuous variable have many pitfalls as pointed out by Giannoni A and co-workers (28). Although we made three times standard deviation of the mean from the controls to avoid least possible false positive results, it does not necessarily provide our study with the correct upper normal limits.

12) In the discussion, I do not understand why authors attribute the loss in MPI and LV strain to underfilling also in patients with completely normal pulmonary artery pressure. Why should those patients have a problem of the underfilling? Are there actually any significant differences in pulmonary vascular resistances in your population? From the data you showed patients with COPD and normal pulmonary pressure have slightly higher LV systolic volumes as compared to controls (underfilling?) Is there any relationship between PVR and MPI or LV strain? Please avoid overstatement and keep the discussion closer to the data you have. The problem of inflammation in this respect seems more promising and supported by data.

Answer: This part requires are little more space:

First: “patients with completely normal pulmonary artery pressure.” Our no PH (no pulmonary hypertension) had 18±3 and those with PH had 29±4 mmHg in mean pulmonary pressure (mPAP) (Normal mPAP is 14±3 mmHg,) which reflect resting state. We have, however, in another invasive study, in the same patients by right heart catheterisation, demonstrated that even those with no PH have a pathological rise in their pulmonary pressure when they exercise (10).

Second: “Are there actually any significant differences in pulmonary vascular resistances in your population?” Again: According to the existing upper normal limits at that time (2012), PVR was elevated at rest in all patients with PH and in 50 patients (69%) in no-PH group (10).

Third: Under-filling: Our data show that left atrial and LV size were undersized compared to the right side. We consider this, together with the impaired RV function and the significant associations between stroke volume and left atrium volume (decreasing SV and decreasing left atrium), is a reflection of under-filling of the left side. This is well illustrated in figure 2B and 2C where a gradual reduction of stroke volume is accompanied with a decrease in MPI and strain. Under-filling of the left side in similar patients has also been emphasized by others (See ref 1-3 in the article) However, we acknowledge that we have presented this more or less as the only explanation, which we regret. We have thus modified this by adding the sentence below on page 14, first paragraph. We are grateful to the reviewer who has made us aware of this.

Page 14, paragraph one: Although under-filling probably is not the only mechanism for the reduced size of the left side, 

13) Please give more relevance also in the results and the discussion to the decrease observed in 2D and 3D LVEF and RVEF, do not focus only on MPI and LV strain, this seems a biased way of presenting your data.

Answer: We agree with the reviewer that the findings regarding 2D and 3D EF are too little focused in the article, and we have therefore added a supplement in the discussion part on page 11, third paragraph (See below). However, the right heart function in general including RVEF, has been thoroughly featured in our previous JACC paper, which is also referred to several times (11). Thus, we do not find it correct to go deeper into this topic in the present paper. 

Page 11, third paragraph: Moreover, traditional method as 2D and novel 3D ejection fraction were also reduced in COPD with and without PH as compared to normal individuals. The difference, however, was small and probably not of clinical significance, but emphasizes the findings of LV strain and MPI. In addition, LV ejection fraction by CMR was also found mildly reduced in 19%. 

References

1. Ingunn Skjørten , Janne Mykland Hilde, Morten Nissen Melsom, Viggo Hansteen, Kjetil Steine, Sjur Humerfelt Pulmonary Artery Pressure and PaO2 in Chronic Obstructive Pulmonary Disease. Respir Med. 2013;107:1271-9.

2. Cazzola M. Bettoncelli G. Sessa E. Cricelli C. Biscione G. Prevalence of comorbidities in patients with chronic obstructive pulmonary disease. Respiration. 2010; 80: 112-119 

3. Hong Y, Graham MM, Southern D, McMurtry MS. The Association between Chronic Obstructive Pulmonary Disease and Coronary Artery Disease in Patients Undergoing Coronary Angiography.COPD. 2019;16:66-71.

4. Fuller T, Movahed A. Current review of exercise testing: application and interpretation. Clin Cardiol.1987;10:189-200.

5. Chew SK, Colville D, Canty P, Hutchinson A, Wong A, Luong V, Wong TY, McDonald C, Savige J. Kidney Hypertensive/Microvascular Disease and COPD: a Case Control Study. Blood Press Res. 2016;41:29-39.

6. Paulus WJ, Tschöpe C. A novel paradigm for heart failure with preserved ejection fraction: comorbidities drive myocardial dysfunction and remodeling through coronary microvascular endothelial inflammation. J Am Coll Cardiol. 2013;62:263-71.

7. Aagaard EN, Kvisvik B, Pervez MO, Lyngbakken MN, Berge T, Enger S, Orstad EB, Smith P, Omland T, Tveit A, Røsjø H, Steine K.Left ventricular mechanical dispersion in a general population: Data from the Akershus Cardiac Examination 1950 study. Eur Heart J Cardiovasc Imaging. 2020;21:183-190.

8. Matteo Cameli, Carlotta Sciaccaluga, Ferdinando Loiacono, Iana Simova, Marcelo H Miglioranza, Dan Nistor, Francesco Bandera, Michele Emdin, Alberto Giannoni, Marco M Ciccone, Fiorella Devito, Andrea Igoren Guaricci, Stefano Favale, Matteo Lisi, Giulia E Mandoli, Michael Henein, Sergio Mondillo. The Analysis of Left Atrial Function Predicts the Severity of Functional Impairment in Chronic Heart Failure: The FLASH Multicenter Study. Int J cardiol 2019;286:87-91

9. Alberto Giannoni, Resham Baruah, Tora Leong, Michaela B Rehman, Luigi Emilio Pastormerlo, Frank E Harrell, Andrew J S Coats, Darrel P Francis. Do Optimal Prognostic Thresholds in Continuous Physiological Variables Really Exist? Analysis of Origin of Apparent Thresholds, With Systematic Review for Peak Oxygen Consumption, Ejection Fraction and BNP. PLOSone 2014;9:

10. Janne Mykland Hilde, Ingunn Skjørten, Ole Jørgen Grøtta, Viggo Hansteen, Morten Nissen Melsom, Jonny Hisdal, Sjur Humerfelt, Kjetil Steine. Haemodynamic responses to exercise in patients with COPD. Eur Respir J 2013;41:1031–41.

11. Janne Mykland Hilde, Ingunn Skjørten, Ole Jørgen Grøtta, Viggo Hansteen, Morten Nissen Melsom, Jonny Hisdal, Sjur Humerfelt, Kjetil Steine, Right Ventricular Dysfunction and Remodeling in Chronic Obstructive Pulmonary Disease Without Pulmonary Hypertension. Journal of the American College of Cardiology 2013;62:1103-1111.

Answers to Reviewer #2 concerning the article

Left ventricular dysfunction in COPD without pulmonary hypertension

Reviewer #2: I have read with great interest the article regarding LV dysfunction in COPD patients without pulmonary hypertension. The novelty in research paper is slightly dubious. Researches have already studied this problem. For example- Willem-Jan Flu A et al (Co-existence of COPD and left ventricular dysfunction in vascular surgery patients. Respiratory Medicine (2010) 104, 690-696) have shown that mild COPD and subclinical LV dysfunction often coexist, the combination of these two is associated with an increased risk for long-term all-cause mortality. Also authors have mentioned about using preoperative ‘integrated cardiopulmonary risk index’ by performing standard preoperative spirometry and echocardiography. 

Answer: Willem-Jan Flu A and co-workers have studied 1003 patients undergoing prospectively peripheral vascular or endovascular surgery (1). This and the present study are, however, quite different of two main reasons: First, according to a review article by Gersh BJ et al., the prevalence of serious angiographic coronary artery disease ranges from 37% to 78% in patients undergoing operation for peripheral vascular disease (2). The LV dysfunction in the study by Willem-Jan Flu A was therefore probably mainly caused by ischemic heart disease (1). Second, the aim of the present study was to find out if the COPD disease itself has a negative impact on LV function. It was therefore a main issue to exclude patients with coronary heart disease, since this is also present in COPD. This was performed by a systematic examination by one of the authors (IS), where approximately 1300 outpatients COPD were screened and then examined by a bicycle exercise test (JMH) and echo (JMH), and patients with LV EF < 50% were excluded. We cannot therefore agree with reviewer #2 that “The novelty in research paper is slightly dubious”, and we hope that what is written above is elucidating in this concern. We are, however, aware, even after such a thorough selecting process, that a few patients with silent coronary heart disease may have been included in our study. We believe, however, that this has not had any important impact on our result.

Current study confirms abovementioned findings about coexisting of COPD and subclinical LV dysfunction, but also adds mechanisms for the reduced LV systolic function in COPD patients without pulmonary hypertension. Researchers have found that LV MPI and strain are two simple echo indices of LV global and systolic function in these patients and based on this they recommend to implement it in the examination of patients with COPD. In the light of the above; this study may be helpful in COPD management and can be used as simplified COPD management tool.

Answer: We are thankful for the recognition of our work regarding the usefulness of LV MPI and LV strain in the assessment of LV function in patients with COPD. 

References

1. Willem-Jan Flu, Yvette R B M van Gestel, Jan-Peter van Kuijk, Sanne E Hoeks, Ruud Kuiper, Hence J M Verhagen, Jeroen J Bax, Don D Sin, Don Poldermans. Co-existence of COPD and Left Ventricular Dysfunction in Vascular Surgery Patients. Respir med 2010;104:690-96.

2. Gersh BJ, Rihal CS, Rooke TW, Ballard DJ. Evaluation and management of patients with both peripheral vascular and coronary artery disease. J Am Coll Cardiol. 1991;18:203-14.

8

---

## [Decision Letter · Decision Letter 1]

9 Jun 2020

Left ventricular dysfunction in COPD without pulmonary hypertension

PONE-D-20-04365R1

Dear Dr. Steine,

We’re pleased to inform you that your manuscript has been judged scientifically suitable for publication and will be formally accepted for publication once it meets all outstanding technical requirements.

Kind regards,

Vincenzo Lionetti, M.D., PhD

Academic Editor

PLOS ONE

Additional Editor Comments (optional):

Reviewers' comments:

Reviewer's Responses to Questions

**Comments to the Author**

1. If the authors have adequately addressed your comments raised in a previous round of review and you feel that this manuscript is now acceptable for publication, you may indicate that here to bypass the “Comments to the Author” section, enter your conflict of interest statement in the “Confidential to Editor” section, and submit your "Accept" recommendation.

Reviewer #1: All comments have been addressed

Reviewer #2: (No Response)

2. Is the manuscript technically sound, and do the data support the conclusions?

Reviewer #1: Yes

Reviewer #2: Yes

3. Has the statistical analysis been performed appropriately and rigorously? 

Reviewer #1: Yes

Reviewer #2: Yes

4. Have the authors made all data underlying the findings in their manuscript fully available?

Reviewer #1: No

Reviewer #2: Yes

5. Is the manuscript presented in an intelligible fashion and written in standard English?

Reviewer #1: Yes

Reviewer #2: Yes

6. Review Comments to the Author

Reviewer #1: The authors should be awarded for their effort to go through all my previous concerns. I believe that the paper now deserves to be published.

Reviewer #2: The novelty in research paper is slightly dubious. Researches have already studied this problem. For example- Willem-Jan Flu A et al (Co-existence of COPD and left ventricular dysfunction in vascular surgery patients. Respiratory Medicine (2010) 104, 690-696) have shown that mild COPD and subclinical LV dysfunction often coexist, the combination of these two is associated with an increased risk for long-term all-cause mortality. Also authors have mentioned about using preoperative ‘integrated cardiopulmonary risk index’ by performing standard preoperative spirometry and echocardiography. Current study confirms abovementioned findings about coexisting of COPD and subclinical LV dysfunction, but also adds mechanisms for the reduced LV systolic function in COPD patients without pulmonary hypertension. Researchers have found that LV MPI and strain are two simple echo indices of LV global and systolic function in these patients and based on this they recommend to implement it in the examination of patients with COPD. In the light of the above this study may be helpful in COPD management and can be used as simplified COPD management tool. Reviewer suggests accepting the article.

7. PLOS authors have the option to publish the peer review history of their article (what does this mean?). If published, this will include your full peer review and any attached files.

Reviewer #1: No

Reviewer #2: No

---

## [Editor Report · Acceptance letter]

30 Jun 2020

PONE-D-20-04365R1 

Left ventricular dysfunction in COPD without pulmonary hypertension 

Dear Dr. Steine:

I'm pleased to inform you that your manuscript has been deemed suitable for publication in PLOS ONE. Congratulations! Your manuscript is now with our production department. 

Kind regards, 

on behalf of

Prof. Vincenzo Lionetti 

Academic Editor

PLOS ONE